# Evolutionarily Conserved Role of Thioredoxin Systems in Determining Longevity

**DOI:** 10.3390/antiox12040944

**Published:** 2023-04-17

**Authors:** Abdelrahman AlOkda, Jeremy M. Van Raamsdonk

**Affiliations:** 1Department of Neurology and Neurosurgery, McGill University, Montreal, QC H3A 2B4, Canada; 2Metabolic Disorders and Complications Program, Research Institute of the McGill University Health Centre, Montreal, QC H4A 3J1, Canada; 3Brain Repair and Integrative Neuroscience Program, Research Institute of the McGill University Health Centre, Montreal, QC H4A 3J1, Canada; 4Division of Experimental Medicine, Department of Medicine, McGill University, Montreal, QC H4A 3J1, Canada

**Keywords:** aging, lifespan, thioredoxin, reactive oxygen species, redox signaling, animal models, *C. elegans*, *Drosophila*, mouse models, genetics

## Abstract

Thioredoxin and thioredoxin reductase are evolutionarily conserved antioxidant enzymes that protect organisms from oxidative stress. These proteins also play roles in redox signaling and can act as a redox-independent cellular chaperone. In most organisms, there is a cytoplasmic and mitochondrial thioredoxin system. A number of studies have examined the role of thioredoxin and thioredoxin reductase in determining longevity. Disruption of either thioredoxin or thioredoxin reductase is sufficient to shorten lifespan in model organisms including yeast, worms, flies and mice, thereby indicating conservation across species. Similarly, increasing the expression of thioredoxin or thioredoxin reductase can extend longevity in multiple model organisms. In humans, there is an association between a specific genetic variant of thioredoxin reductase and lifespan. Overall, the cytoplasmic and mitochondrial thioredoxin systems are both important for longevity.

## 1. Introduction

Aging is an intrinsic process that causes a progressive loss of function over time that increases the probability of death. While the aging process remains incompletely understood, research in multiple model organisms has begun to unravel the molecular mechanisms involved. In yeast, worms, flies and mice, modulating the expression of a single gene out of thousands of genes is sufficient to affect lifespan, thereby providing insight into the genetic pathways that determine longevity.

The Free Radical Theory of Aging proposes that aging results primarily from the accumulation of oxidative damage caused by reactive oxygen species (ROS) [1]. ROS are highly reactive oxygen-containing molecules that can damage cellular components, including DNA, proteins, and lipids. ROS are generated during normal cellular metabolism, but their levels can be increased through exposure to environmental stressors or internal stressors such as inflammation or metabolic dysfunction. In order to detoxify ROS and repair ROS-mediated damage, organisms have evolved to express antioxidant enzymes, including thioredoxin (TRX/TXN) and thioredoxin reductase (TRXR/TXNRD). 

Thioredoxin and thioredoxin reductase combine to form thioredoxin systems in different compartments of the cell and play a crucial role in regulating cellular redox homeostasis [2]. The thioredoxin system acts to reduce proteins, both to repair oxidative stress and modulate their activity. Thioredoxin also plays an important role in intracellular signaling. In this review, we discuss the different roles of thioredoxin and thioredoxin reductase in the cell and how these proteins affect longevity in different model organisms. 

## 2. Antioxidant Roles of Thioredoxin and Thioredoxin Reductase

The thioredoxin system is a crucial redox regulatory system consisting of thioredoxin and its reducing partner thioredoxin reductase, which uses nicotinamide adenine dinucleotide phosphate (NADPH) as an electron donor to reduce thioredoxin (Figure 1). Thioredoxin possesses a unique tertiary structure composed of five β-strands forming the internal core of the protein, four α-helices and a short stretch of helix surrounding the central β-sheets [3] (Figure 2). The active site disulfide is located after the β2-sheet and forms the N-terminal portion of α2. A cis-proline located in a loop preceding β-strand 4 is crucial for the stability and function of thioredoxin. Thioredoxins are the main protein disulfide reductases in the cell and act as electron donors for enzymes via the reversible oxidation of two cysteine thiol groups (-CGPC-, also called CXXC motif, thioredoxin motif and thioredoxin fold) to a disulfide, which is crucial in the thiol-dependent antioxidant system [4]. The thioredoxin fold structure of thioredoxin is shared among a group of proteins that serve as key players in redox signaling and control, all of which can act to reduce disulfides. These proteins include glutaredoxin (GRX), glutathione peroxidase (GPX), glutathione transferase (GST), thioredoxin peroxidase (also known as peroxiredoxin or PRX) and protein disulfide isomerase (PDI) [5,6].

The main antioxidant function of thioredoxin involves the transfer of two electrons and two protons, resulting in the covalent interconversion of a disulfide and a dithiol. During this reaction, cysteines located at positions 32 and 35 of thioredoxin execute a bimolecular nucleophilic substitution mechanism to transfer electrons from thioredoxin to the substrate protein [5]. Firstly, the N-terminal cysteine of thioredoxin initiates a nucleophilic attack on the substrate protein’s disulfide bond, leading to the formation of a mixed disulfide bond between thioredoxin and the substrate protein (Figure 3). Subsequently, the C-terminal cysteine of thioredoxin initiates a nucleophilic attack on the intermediate intermolecular disulfide bond, forming a disulfide bond in the oxidized thioredoxin and breaking the disulfide bond in the reduced substrate protein. 

Thioredoxin is maintained in its active and reduced form primarily by thioredoxin reductase but can also be reactivated by glutaredoxin in the glutathione (GSH) system [9,10]. Thioredoxin can act as an antioxidant either directly by quenching singlet oxygen and scavenging of hydroxyl radicals or indirectly by reducing proteins oxidized by ROS [11]. One of the most important targets of thioredoxin is peroxiredoxin, which acts to directly reduce peroxides such as H_2_O_2_ and various alkyl hydroperoxides [12,13]. Once peroxiredoxin reduces its target, thioredoxin restores peroxiredoxin activity by recycling the oxidized form of peroxiredoxin back to its reduced state.

Thioredoxin reductase is an oxidoreductase that uses NADPH to reduce the active-site disulfide of thioredoxin, thereby restoring thioredoxin’s activity [14]. Thioredoxin reductase contains flavin adenine dinucleotide (FAD) and pyridine nucleotide disulfide. Thioredoxin reductase exists as an antiparallel homodimer with both subunits playing a crucial role in the normal redox reaction during the catalytic cycle. Unlike bacteria and archaea, the active site of thioredoxin reductase in mammalian and multicellular eukaryotes comprises a conserved selenocysteine (Sec) that replaces the Cys_2_ residue located at the penultimate C-terminal position in its X-Cys_1_-Cys_2_-X motif (X is usually Gly or Ser), which is essential for its catalytic function [15]. This substitution confers several advantages including the superior nucleophilicity of Sec, which arises from its ionization under physiological conditions compared to the protonated Cys [16]. Additionally, the position of Sec in the C-terminus provides conformational flexibility that enables it to function as a cellular redox sensor [17].

The first step of the reductive half-reaction of the enzyme involves reduction of the enzyme-bound flavine adenine dinucleotide by NADPH in one subunit [14,18]. From there, the reducing equivalents are transferred to the Cys-Val-Asn-Val-Gly-Cys active site motif of the same subunit, forming a dithiol motif. This dithiol motif reduces the C-terminal selenenyl sulfide motif of the other subunit of the dimer, forming a dithiol or selenolthiol motif [14]. This reduced motif can then reduce the substrates of thioredoxin reductase, including the active site disulfide between positions 32 and 35 of thioredoxin, glutaredoxin 2 (GRX2), PDI, thioredoxin-like-1, granulysin [19,20] and some small molecule substrates such as selenite [21], dehydroascorbate [22], lipoic acid [23], ubiquinone [24], cytochrome C [25] or the cancer drugs motexafin gadolinium [26] and alloxan [27]. Thioredoxin reductase can function as an antioxidant given that it provides electrons to small molecules that can react directly with H_2_O_2_ [20,28]. Thus, the thioredoxin system is an essential redox regulatory system that interacts and collaborates with the glutathione system to maintain the redox balance and protect against oxidative stress in the organism.

## 3. Additional Roles of Thioredoxin and Thioredoxin Reductase: Redox Signaling

In addition to its roles in antioxidant defense, thioredoxin also affects metabolism and intracellular signaling by regulating protein activity (Figure 4). In transferring reducing equivalents from NADPH to target proteins, thioredoxin can modulate their function, structure or stability [29]. The modulation of enzyme activity can result in the binding of substrates or allosteric effectors, leading to metabolic changes. In addition, thioredoxin can activate several transcription factors through redox regulation by modulating their DNA-binding activities. 

Under basal conditions, the redox-regulated apoptosis-signal kinase (ASK1), a member of the MAPKKK family, is directly bound to TXN and TXN2 to maintain low levels of ROS while the thioredoxin-interacting protein TXNIP resides in the nucleus. However, in response to oxidative stress, TXNIP translocates to the cytoplasm and mitochondria and disrupts the binding of TXN-ASK1 and TXN2-ASK1, respectively [30,31]. The disruption can also occur through increased ROS and lead to an overall ROS buildup, mitochondrial distress signaling and eventually an apoptotic signaling cascade. The cascade begins with the phosphorylation of unbound ASK1, leading to the release of cytochrome C and cleavage of caspase-3, initiating downstream apoptotic signaling. 

Additionally, TXNIP inhibits TXN2 protection of mitochondria against ROS [32], leading to mtDNA oxidation and binding of the NOD-like receptor protein 3 (NLRP3) inflammasome [33], ultimately activating the inflammasome [34]. In addition, TXNIP expression is induced by the endoplasmic reticulum unfolded protein response (ER^UPR^) under the IRE1α and PERK-eIF2α pathways [35]. This activation results in the cleavage of pro-interleukin-1β to its active, mature form by caspase-1 and its subsequent production and secretion. Overall, the interaction between TXNIP and thioredoxin plays a crucial role in regulating cellular responses to oxidative stress, including apoptosis and inflammation [36].

Thioredoxin can directly reduce some transcription factors to negatively regulate apoptosis. For example, thioredoxin can activate the nuclear factor (NF)-κB, which regulates the expression of genes that antagonize cell death [37]. Under normal conditions, thioredoxin scavenges ROS in the cytoplasm and inhibits the degradation of IκB. However, increased ROS mediates the degradation of IκB and the nuclear translocation of NF-κB [38]. Nuclear thioredoxin then directly reduces a cysteine of NF-κB and allows NF-κB-dependent gene expression [39]. The role of thioredoxin in inhibiting the ASK-1 and NF-κB pathways suggests that ROS-induced apoptosis may serve as a protective mechanism against chronic oxidative stress.

Thioredoxin can indirectly activate transcription factors responsible for promoting cell viability in response to adverse conditions such as oxidative stress and hypoxia [6]. The DNA-binding activity of these transcription factors is regulated by specific Cys residues, which are reduced by reducing redox factor-1 (Ref-1) as an intermediate in the nucleus. In order for Ref-1 to catalyze this reduction, it needs to be in its reduced form, which is catalyzed by thioredoxin [40]. In addition, Ref-1 is a DNA-repair endonuclease that is involved in the base excision repair (BER) pathway, which is responsible for repair of apurinic/apyrimidinic sites in DNA caused by ROS [41]. Notably, expression of Ref-1 is upregulated in response to oxidative stress [42]. One example of a transcription complex dependent on thioredoxin/Ref-1 interaction is Activator protein-1 (AP-1), the basic region-leucine zipper (bZIP) family of Jun and Fos [43]. Thioredoxin reduces AP-1 cysteines indirectly via Ref-1 and thereby increases the DNA-binding activity of AP-1 to regulate cell growth, differentiation and apoptosis. Another example of thioredoxin’s role in regulating cellular responses is the reduction of a single cysteine residue of the hypoxia-inducible factor 1α (HIF-1α) subunit by thioredoxin/Ref-1 during hypoxia [44,45]. This redox modification is essential for HIF-1 binding with CBP/p300 co-activator to initiate the hypoxic response element (HRE) target genes expression [45]. Like in the previous example, thioredoxin and thioredoxin reductase inhibitors have been shown to downregulate expression of HIF-1α and its subsequent activity [46]. Moreover, studies have demonstrated that thioredoxin and related redox proteins are upregulated in response to hypoxia [47,48], which further emphasizes thioredoxin’s role in HIF-1α regulation. Therefore, thioredoxin possesses multiple important functions in protecting cells from both oxidative stress and the hypoxic-stress response via Ref-1.

Overall, thioredoxin plays an important role in redox signaling in diverse cellular processes, including cell proliferation, differentiation, apoptosis and responses to oxidative stress. 

## 4. Regulation of the Thioredoxin System

The activity of the thioredoxin system is regulated at multiple levels, including gene expression, post-translational modifications, and protein–protein interactions (Figure 5). These regulatory mechanisms allow the system to respond to changes in the cellular redox environment and adapt to different physiological or pathological conditions. 

The thioredoxin system is regulated at the gene expression level by transcription factors, including the nuclear factor erythroid 2–related factor 2 (Nrf2), TATA-binding protein (TBP) and cAMP response element-binding protein (CREB) [49,50]. These transcription factors bind to specific cis-regulatory elements located in the promoter regions of the genes that encode thioredoxin and thioredoxin reductase and are activated in response to various stressors, including oxidative stress and inflammation. This activation induces the expression of the thioredoxin system components, which form part of the cellular stress response. For instance, oxidative stress triggers Nrf2 binding to the antioxidant response element (ARE) present in the thioredoxin promoter [51]. Similarly, the thioredoxin reductase and peroxiredoxin promoters also contain ARE elements that mediate upregulation of their expression in response to oxidative stress [50]. Notably, the reduced form of thioredoxin enhances the binding of Nrf2 to ARE by reducing conserved cysteine residues in the DNA-binding domains of small Maf proteins (sMaf), thereby activating Nrf2-transcription [51,52]. As a result, oxidative stress leads to an increase in thioredoxin levels, which in turn activates the transcription factors responsible for inducing even higher levels of thioredoxin and other antioxidants. 

Post-translational modifications, such as phosphorylation, acetylation or S-nitrosylation, can also modulate the activity of the thioredoxin system. For example, phosphorylation of thioredoxin at specific residues has been shown to increase its activity or alter its substrate specificity [53], while S-nitrosylation [54] and glutathionylation [55] can impair its function by interfering with the formation of disulfide bonds.

Protein–protein interactions are also important for the regulation of the thioredoxin system. For instance, TXNIP, also known as thioredoxin binding protein-2 (TBP-2), is an important endogenous molecule that negatively regulates the function of thioredoxin [56,57]. There are two mechanisms through which TXNIP inhibits thioredoxin function and activity. Firstly, TXNIP competes with thioredoxin for binding sites, which removes thioredoxin from proteins that are inhibited by the steric effect of TXN1 binding, such as redox-regulated apoptosis-signal kinase 1 (ASK1) [30]. TXNIP inhibits thioredoxin activity in a redox-dependent manner by forming a mixed disulfide bond with reduced thioredoxin active site thiols through thioredoxin active site Cys32 and TXNIP Cys247 [57]. Secondly, the increased and overexpressed TXNIP, as seen in response to factors such as disturbed flow and high glucose [58,59], results in a reduction in the thioredoxin system activity. Thus, the increased formation of TXNIP-TXN complexes leads to a higher concentration of oxidized proteins when exposed to oxidative stress.

In addition to these intrinsic regulatory mechanisms, the thioredoxin system can also be modulated by extrinsic factors such as dietary interventions, drugs or environmental stressors. For example, several dietary compounds, such as resveratrol [60], curcumin [61,62] or sulforaphane [63], have been shown to modulate the expression and the activity of thioredoxin and other antioxidant enzymes. Similarly, some drugs, such as statins [64,65] or angiotensin receptor blockers [66,67], have been reported to enhance the activity of the thioredoxin system and reduce oxidative stress.

## 5. Cytoplasmic Thioredoxin System Contributes to Lifespan in Yeast

The thioredoxin system is evolutionarily conserved with multiple forms distributed in different compartments of the cell [68]. Yeast possesses a cytoplasmic thioredoxin system consisting of the thioredoxin TRX1 and thioredoxin reductase TRR1 and a mitochondrial thioredoxin system that includes the thioredoxin TRX3 and thioredoxin reductase TRR2 (Appendix A). In yeast, replicative lifespan is measured as the number of cell divisions that a mother cell can undertake to produce daughter cells. Chronologic lifespan is the length of time that a yeast cell maintains the ability to generate new colonies in a non-dividing state, for example, when a specific cell density is reached in liquid culture. Disruption of the cytoplasmic thioredoxin gene *TRX1* results in decreased chronologic lifespan [69,70], while disruption of the mitochondrial thioredoxin gene *TRX3* has no effect on chronologic or replicative lifespan [71,72] (Table 1). Similarly, disruption of the cytoplasmic thioredoxin reductase gene *TRR1* reduces chronologic lifespan [73], while loss of the mitochondrial thioredoxin reductase gene *TRR2* does not affect chronologic or replicative lifespan [71,72]. Together, this indicates that the cytoplasmic thioredoxin system is required for normal longevity in yeast, while the mitochondrial thioredoxin system is dispensable.

## 6. Cytoplasmic Thioredoxin Is Important for Longevity in *Caenorhabditis elegans*


In *C. elegans,* there are at least five different thioredoxins (TRX-1, TRX-2, TRX-3, TRX-4 and TRX-5) and two thioredoxin reductases (TRXR-1 and TRXR-2). TRX-1 and TRXR-1 make up the cytoplasmic thioredoxin system, while TRX-2 and TRXR-2 form the mitochondrial thioredoxin system (Appendix A). In *C. elegans,* disruption of the cytoplasmic thioredoxin gene *trx-1* results in a clear decrease in lifespan in wild-type worms and multiple long-lived mutant strains [74,75,76,77] (Table 1). Deletion of *trx-1* also results in decreased resistance to exogenous stressors and elevated levels of reactive oxygen species [74,75]. The large effect of *trx-1* disruption on lifespan is perhaps surprising given that its expression is primarily limited to a small number of neurons (ASI and ASJ) and part of the intestine [75,76] and accounts for less than 1% of the total thioredoxin mRNA [74]. A role for *trx-1* in determining lifespan is supported by the observation that overexpression of *trx-1* is sufficient to extend longevity [77]. In contrast to *trx-1,* disruption of the cytoplasmic thioredoxin reductase gene *trxr-1* does not affect lifespan [74,78,79], and *trxr-1* mutants do not exhibit decreased survival after exposure to oxidative stress or other exogenous stressors [74,78,79]. Together, this suggests that *trx-1* performs functions in the cell that are independent of *trxr-1* and are important for lifespan and cellular resilience. It is possible that *trx-1* can be re-activated by another enzyme or that redox-independent functions of *trx-1* are important for stress resistance and longevity. 

As in yeast, deletion of the mitochondrial thioredoxin gene *trx-2* does not affect lifespan in wild-type worms [74,80], though it is required for lifespan extension in the long-lived mitochondrial mutants *nuo-6* and *isp-1* [74]. Disruption of *trx-2* also has little or no detrimental effect on resistance to oxidative or other stresses in wild-type animals [74,80]. The lifespan of *trxr-2* mitochondrial thioredoxin reductase mutants is equivalent to or slightly decreased compared to wild-type worms, depending on the precise experimental conditions [74,79,80]. Similar to *trx-2, trxr-2* is specifically required for the extended longevity of *nuo-6* and *isp-1* mutants [74] but not long-lived *daf-2* insulin/IGF-1 receptor mutants [80]. The loss of *trxr-2* does not decrease resistance to oxidative or other stresses [74,80], despite resulting in increased levels of ROS [74,79]. 

Deletion of *trx-3* does not affect lifespan or resistance to stress, while *trx-3* overexpression provides modest protection against exposure to bacterial pathogens [81]. Overall, cytoplasmic thioredoxin is required for both lifespan and resistance to stress in *C. elegans,* while cytoplasmic thioredoxin reductase and both components of the mitochondrial thioredoxin system are dispensable. Although *trx-1* is required to achieve a normal lifespan in *C. elegans*, this gene is not essential, as *trx-1* mutants are viable, fertile and develop to adulthood. 

## 7. Contribution of Thioredoxin Systems to Lifespan in *Drosophila*


*Drosophila melanogaster* possess a male-specific thioredoxin, thioredoxinT (TrxT), and a female specific thioredoxin, Deadhead (Dhd) and both of these are present in the nucleus in the germline. *Drosophila* also possess thioredoxin 2 (Trx-2), which is present in the nucleus. There are two thioredoxin reductases in *Drosophila*: Trxr-1 possess isoforms present in both the cytoplasm and mitochondria, while Trxr-2 is expressed in the mitochondria (Appendix A). Disruption of either the male-specific thioredoxinT (*TrxT*) or female-specific deadhead (*dhd*) thioredoxin genes does not affect longevity in *Drosophila* [82] (Table 1). Overexpression of *TrxT* in all neurons results in increased lifespan and enhanced resistance to oxidative stress [83]. Disruption of the thioredoxin gene *Trx-2* decreases lifespan [82,84], while overexpression of this gene extends longevity [82]. *Trx-2* mutants have been shown to possess increased resistance to hydrogen-peroxide-mediated oxidative stress [82] but decreased resistance to paraquat-mediated oxidative stress [84]. *Trx-2* overexpression flies show increased resistance to both types of oxidative stress [82,84], while flies lacking all three thioredoxin genes (*TrxT, dhd* and *Trx-2*) show decreased resistance to hydrogen-peroxide-mediated oxidative stress compared to wild-type flies [82].
antioxidants-12-00944-t001_Table 1Table 1Effect of thioredoxin systems on lifespan in model organisms.OrganismGeneLocationModulationEffect on Lifespan**References**Yeast*TRX1*CytoplasmDisruption↓[69,70]*TRR1*CytoplasmDisruption↓[73]*TRX3*MitochondriaDisruption=[71,72]*TRR2*MitochondriaDisruption=[71,72]*C. elegans**trx-1*CytoplasmDisruption↓[74,75,76,77]*trx-1::GFP*CytoplasmOverexpression↑[77]*trxr-1*CytoplasmDisruption=[74,78,79]*trx-2*MitochondriaDisruption=[74,80]*trxr-2*MitochondriaDisruption=/↓[74,79,80]*trx-3*Cytoplasm/NucleusIntestineDisruption=[81]*Drosophila**TrxT*Nucleus, male specificDisruption=[82]*TrxT*Nucleus, male specificOverexpression in all neurons↑[83]*dhd*Nucleus, female specificDisruption=[82]*Trx-2*NucleusDisruption↓[82,84]*Trx-2*NucleusOverexpression↑[82]*TrxT-dhd-Trx-2*NucleusDisruption=/↓[82]*Trxr-1*Cytoplasm and MitochondriaDisruptionLarval lethality[85,86]*Trxr-1*Cytoplasm and MitochondriaOverexpression=[87]*Trxr-2*MitochondriaOverexpression↑[87]*Vdup1*Cytoplasm/NucleusDownregulation↑[88]*Vdup1*Cytoplasm/NucleusOverexpression↓[88]Mice*Txn1/Trx1*CytoplasmicDisruptionEmbryonic lethal[89]*Txn1+/−*CytoplasmicHeterozygous disruption=[90,91]Human *TXN*CytoplasmicOverexpression↑[92]Human *TXN*CytoplasmicOverexpression↑ early lifespanin males[93]Human *TXN*CytoplasmicOverexpression=[94]*Txnrd1*CytoplasmicDisruptionEmbryonic lethal[95]*Txn2/Trx2*MitochondriaDisruptionEmbryonic lethal[96]*Txn2+/−*MitochondriaHeterozygous disruption=[97]Human *TXN2*MitochondriaOverexpression↑ early lifespan[98]Human *TXN1* + human *TXN2*Cytoplasmic and MitochondriaOverexpression↓[99]*Txn+/−; Txn2+/−*Cytoplasmic and MitochondriaHeterozygous disruption↑[91]
*Txnrd2*MitochondriaDisruptionEmbryonic lethal[100]RatsHuman *TXN*CytoplasmicOverexpression=[91]“↓” lifespan decreased compared to WT. “↑” lifespan increased compared to WT. “=” lifespan equivalent to WT.


A null mutation in the cytoplasmic thioredoxin reductase gene *Trxr-1* results in larval lethality, while mutations that decrease *Trxr-1* activity markedly reduce adult lifespan [85]. The lifespan deficit in *Trxr-1* mutants is partially rescued by overexpression of catalase, suggesting that diminished oxidative stress defense contributes to the decrease in longevity [85]. Interestingly, *Trxr-1* encodes a cytoplasmic and mitochondrial isoform, both of which affect longevity independently of the other isoform [86]. While overexpression of *Trxr-1* is not sufficient to extend longevity [87], overexpression of the mitochondrial thioredoxin reductase gene *Trxr-2* increases lifespan [87].

The thioredoxin-interacting protein TXNIP is a negative regulator of thioredoxin. Knockdown of *Vdup1,* the *Drosophila* homolog of *TXNIP,* with RNAi increases thioredoxin activity and results in an increase in mean lifespan and a slight increase in resistance to oxidative stress [88]. Overexpression of *Vdup1* results in the opposite effect, decreasing thioredoxin activity, decreasing lifespan and decreasing oxidative stress resistance [88].

Overall, *Drosophila* lifespan is highly dependent on TRXR-1 activity, while TRX-2 modestly contributes to longevity. 

## 8. Cytoplasmic and Mitochondrial Thioredoxin Systems Are Required for Survival in Mice 

In mice, there is a cytoplasmic thioredoxin system consisting of the thioredoxin TXN1 and the thioredoxin reductase TXNRD1 and mitochondrial thioredoxin system consisting of the thioredoxin TXN2 and the thioredoxin reductase TXNRD2. Mice also have a third thioredoxin reductase TXNRD3, which is present in the cytoplasm and nucleus (Appendix A). Mice that are homozygous for the targeted inactivation of the cytoplasmic thioredoxin gene *Txn1/Trx1* are embryonic lethal [89], while heterozygous *Txn1+/−* mice have a wild-type lifespan [90,91] (Table 1). The targeted inactivation of the cytoplasmic thioredoxin reductase gene *Txnrd1/TrxR1* also results in embryonic lethality [95]. Although one study reported that overexpression of human *TXN* under the human β-actin promoter significantly extends longevity in mice [92], a subsequent study using the same mice found that only early life survival in male mice was significantly increased via *TXN* overexpression [93]. This difference may have arisen due to a relatively short wild-type mouse lifespan in the earlier study. Nonetheless, both studies reported beneficial effects of *TXN* overexpression, including protection against focal ischemia [101], increased survival of isolated cells after UV stress [92], decreased oxidative damage [93] and increased resistance to oxidative stress [93]. A third study examined the effect of overexpressing human *TXN* under the endogenous *TXN* promoter, as expression from the β-actin promoter decreases with age, which may have accounted for the lack of effect on maximum lifespan in the second study. While there appeared to be a mild increase in survival at early time points, there was overall no statistically significant difference between *TXN* overexpressing mice and wild-type animals [94].As with the cytoplasmic thioredoxin system, the mitochondrial thioredoxin system is also essential for embryonic development in mice. Targeted inactivation of the mitochondrial thioredoxin gene *Txn2/Trx2* [96] or the mitochondrial thioredoxin reductase gene *Txnrd2/TrxR2* [100] results in embryonic lethality. Mice heterozygous for the *Txn2* gene (*Txn2+/−* mice) possess a wild-type lifespan [102], despite exhibiting elevated levels of ROS and increased oxidative damage to DNA, protein and lipids [97]. Similar to overexpression of *TXN,* transgenic mice expressing increased levels of human *TXN2* under the human endogenous promoter show a small increase in lifespan early in life but no change in maximum lifespan [98].

Interestingly, simultaneous overexpression of human *TXN* and *TXN2* decreases lifespan in mice, despite the overexpression of each gene individually mildly increasing early lifespan [99]. The detrimental effect on longevity was attributed to an increased incidence of cancer. Consistent with this result, mice that are heterozygous for the inactivation of both *Txn1* and *Txn2* exhibit a significant increase in lifespan, which may be due to a decrease in cancer incidence [91]. *Txnrd2*-transgenic mice have recently been generated but their lifespan has yet to be determined [103]. Cells derived from the *Txnrd2*-transgenic mice exhibit increased resistance to oxidative stress. It will be interesting to determine the lifespan of these animals and the extent to which overexpression of *Txnrd1* will affect longevity.

In summary, all components of the cytoplasmic and mitochondrial thioredoxin systems are essential for embryonic development in mouse models. It is unclear whether the lack of survival during embryonic development results from a severe shortening of lifespan or whether functional thioredoxin systems are required for important processes in embryonic development. To examine the effect of thioredoxin-system disruption on lifespan independent of embryonic development, one could generate adult-only knockout animals using a Cre/lox system or use an inducible expression system to express thioredoxin during development in a thioredoxin-knockout background. Results from the overexpression studies indicate that modulation of thioredoxin genes affects longevity in mice. 

## 9. Thioredoxin Reductase Variant Is Associated with Longevity in Humans

Similar to mice, humans possess TXN and TXNRD1 in the cytoplasm and TXN2 and TXNRD2 in the mitochondria with a third thioredoxin reductase TXNRD3 present in the cytoplasm and nucleus (Appendix A). While it is not possible to genetically manipulate the expression levels of thioredoxin system genes in humans, multiple studies have identified genetic variants that are associated with extended longevity. In a study comparing oldest–old individuals (age 92–93) with middle-aged Danes, an allele of the cytoplasmic thioredoxin reductase gene *TXNRD1* was found to be associated with longevity [104]. A subsequent study found that genetic variation in *TXNRD1* is associated with physical and cognitive performance in very old individuals [105]. The association of *TXNRD1* with physical performance in old age was confirmed in a cohort from Southern Italy [106], while the association of *TXNRD1* with longevity was supported by results from a Dutch cohort, which showed the same relationship but failed to reach significance [104]. Taken together, these results suggest that the thioredoxin system may also contribute to longevity in humans. As the number of studies examining the role of the thioredoxin system in longevity in humans is currently limited, additional evidence would strengthen this conclusion.

## 10. Discussion

### 10.1. Relative Importance of Cytoplasmic and Mitochondrial Thioredoxin Systems Differs between Species

The role of the cytoplasmic and mitochondrial thioredoxin systems in determining lifespan has been examined in multiple genetic model organisms through increasing or decreasing the expression of thioredoxin or thioredoxin reductase. While there is evidence for a contribution of the thioredoxin systems to longevity in yeast, worms, flies, mice and humans, the relative importance of each component of these systems varies between species. In yeast and *C. elegans*, disruption of the cytoplasmic thioredoxin system results in the largest detrimental effect on longevity, while disruption of the mitochondrial thioredoxin system has minimal impact on lifespan. In *Drosophila,* both the cytoplasmic and mitochondrial thioredoxin systems affect lifespan, with the largest effect observed with the cytoplasmic thioredoxin reductase. In mice, both the cytoplasmic and mitochondrial thioredoxin systems are essential for life as disruption of any of the components results in embryonic lethality. Thus, it appears that in more-complex organisms, there is a greater reliance on thioredoxin systems for survival and an increased importance of the mitochondrial thioredoxin system compared to less-complex organisms. 

### 10.2. Cytoplasmic and Mitochondrial Thioredoxin Systems Act Independently to Affect Longevity

In reviewing the literature, we found that disrupting the cytoplasmic or mitochondrial thioredoxin systems often resulted in different effects on longevity, especially in yeast and *C. elegans.* This clearly indicates that the cytoplasmic and mitochondrial thioredoxin systems do not perform redundant functions. It is important to possess a functioning thioredoxin system in both of these compartments of the cell. As one of thioredoxin’s main functions is to reduce disulfide bonds in oxidized proteins, thioredoxin is required to reduce proteins in all parts of the cell. Precise subcellular localization is likely also important for thioredoxin’s roles in intracellular signaling and as a molecular chaperone.

### 10.3. Thioredoxin and Thioredoxin Reductase can Affect Lifespan Independently

Although in some cases modulating the expression of the thioredoxin gene resulted in the same effect as modulating the expression of the corresponding thioredoxin reductase gene on lifespan, there were also examples in which different effects were observed. For example, in *C. elegans*, deletion of the cytoplasmic thioredoxin gene *trx-1* decreases lifespan and stress resistance, while loss of the cytoplasmic thioredoxin reductase gene *trxr-1* does not reduce longevity or resistance to stress. In *Drosophila,* disruption of *TrxT* or *dhd* does not affect lifespan, while loss of *Trxr-1* results in larval lethality. In cases where thioredoxin disruption produces a phenotype while disruption of the corresponding thioredoxin reductase does not, this suggests that the thioredoxin possesses a thioredoxin-reductase-independent function or that multiple enzymes can restore thioredoxin activity. As thioredoxins have been shown to possess redox-independent functions [107,108,109,110,111], disruption of thioredoxin and thioredoxin reductase would produce different results if it is a redox-independent function of thioredoxin that is contributing to its effect on lifespan. In cases where the phenotype resulting from disruption of thioredoxin reductase is more severe than disruption of the corresponding thioredoxin, it is possible that the thioredoxin reductase is also acting on other targets, leading to a broader effect than just disrupting the thioredoxin.

### 10.4. The Effect of Thioredoxin and Thioredoxin Reductase on Lifespan Is Correlated with Effect on Resistance to Oxidative Stress

As thioredoxin performs multiple functions within the cell as an antioxidant, signaling molecule and molecular chaperone, it is important to determine the relative contributions of each of these functions to longevity. In general, it has been observed that resistance to oxidative stress is modulated in the same direction as lifespan. Disruption of *TRR1* in yeast, deletion *trx-1* in *C. elegans,* disruption of *Trx-2* in *Drosophila* and overexpression of *Vdup1* in *Drosophila* all result in decreased lifespan and decreased resistance to oxidative stress. Deletion of *trxr-1, trx-2, trxr-2* or *trx-3* in *C. elegans* does not decrease lifespan or oxidative stress resistance. Overexpression of *TrxT* or *Trx-2* or downregulation of *Vdup1* in *Drosophila* increases lifespan and resistance to oxidative stress. Taken together, these results indicate a correlation between resistance to oxidative stress and lifespan and are consistent with the conclusion that one of the mechanisms by which the thioredoxin systems affect longevity is through modulation of resistance to oxidative stress. 

## 11. Conclusions

Overall, this review highlights the importance of both the cytoplasmic and mitochondrial thioredoxin systems in determining lifespan, which is conserved across species. Disruption of thioredoxin or thioredoxin reductase can have detrimental effects on lifespan in yeast, worms, flies and mice. In addition, overexpression of individual thioredoxin or thioredoxin reductase genes is sufficient to extend longevity in worms, flies and mice. In future studies, it will be important to define the precise molecular mechanisms by which each thioredoxin system affects lifespan. 

## Figures and Tables

**Figure 1 antioxidants-12-00944-f001:**
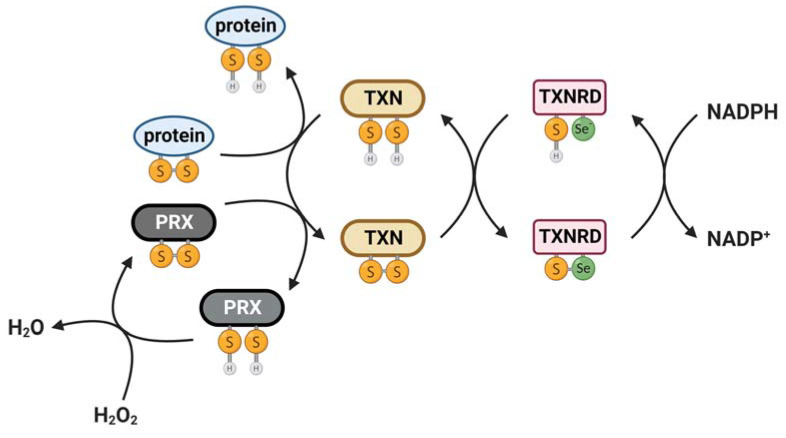
Role of thioredoxin system in antioxidant defense. Thioredoxin (TXN) catalyzes the reduction of disulfides (S-S) within oxidized cellular proteins, which can act to restore protein function. An important target of thioredoxin is the antioxidant peroxiredoxin (PRX), which can detoxify hydrogen peroxide. In reducing target proteins, thioredoxin becomes oxidized. In order to reactivate thioredoxin, thioredoxin reductase (TXNRD) reduces TXN using reducing equivalents obtained from NADPH. S-S = oxidized form. SH = reduced form.

**Figure 2 antioxidants-12-00944-f002:**
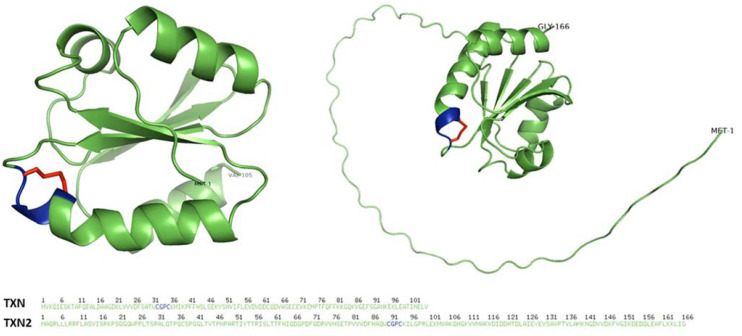
Three-dimensional structure of human thioredoxins. The 3D structure of human cytoplasmic thioredoxin (TXN) (**left**) and mitochondrial thioredoxin 2 (TXN2) (**right**), in their oxidized forms, are shown (UniProt accession P10599 and Q99757, respectively). The 3D representation includes labeled N- and C-termini and highlights the active site thioredoxin fold in blue and the disulfide bond in red. The structures were generated using PyMOL (http://www.pymol.org/pymol, accessed on 31 March 2023) after retrieving TXN1 and TXN2 protein data bank (PDB) formats from AlphaFold [7,8].

**Figure 3 antioxidants-12-00944-f003:**
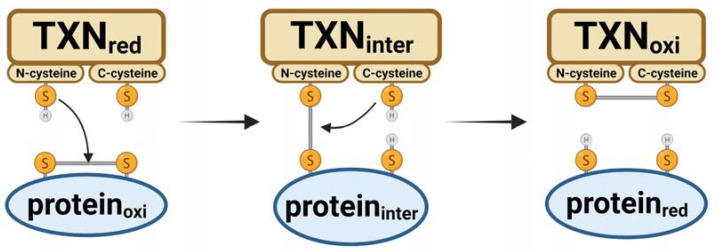
Mechanism of thioredoxin reaction to reduce oxidized proteins. The reduced (red) form of thioredoxin (TXN) can transfer two electrons and two protons to convert an oxidized protein’s disulfide to a dithiol. This reaction involves cysteines at positions 32 (N-terminus) and 35 (C-terminus) of thioredoxin that execute a bimolecular nucleophilic substitution to transfer electrons to the substrate protein. The process ends with an oxidized (oxi) thioredoxin. Inter = intermediate reactant. S-S = oxidized form. SH = reduced form.

**Figure 4 antioxidants-12-00944-f004:**
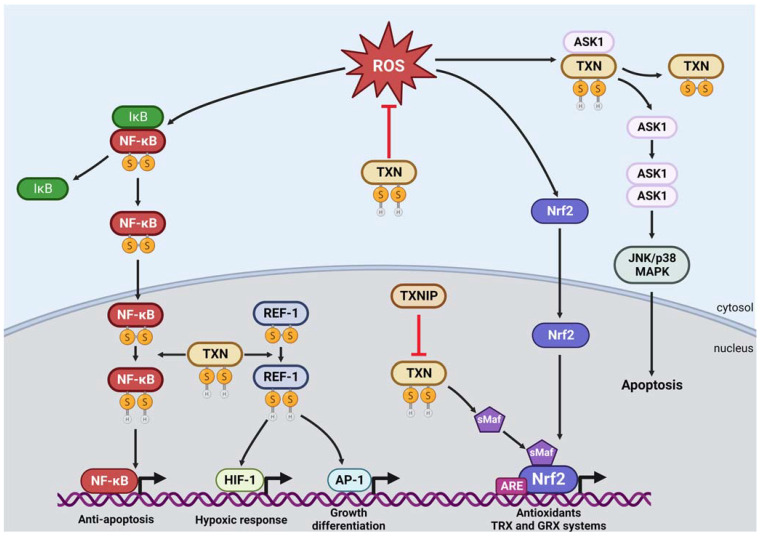
Role of thioredoxin in redox signaling. Thioredoxin (TXN) negatively regulates apoptosis via redox regulation of ASK-1 and inhibition of Iκβ degradation by scavenging ROS in the cytoplasm. In the nucleus, TXN increases the DNA-binding activity of NF-κβ and can enhance the binding of Nrf2 to the antioxidant response element (ARE) through small Maf proteins (sMaf) via reduction of their cysteine residues. TXN also increases the DNA-binding activity of other transcription factors, such as AP-1 and HIF-1, indirectly via the reduction of intermediate Ref-1 cysteine residues. Thioredoxin-interacting protein (TXNIP) can inhibit TXN function by forming a mixed disulfide bond with its reduced form. Note that although thioredoxin reductase is not depicted in this figure, it is important for the reduction of thioredoxin into its reduced, active form. It is the reduced, active form of thioredoxin that contributes to redox signaling. Red lines indicate an inhibitory effect. S-S = oxidized form. SH = reduced form.

**Figure 5 antioxidants-12-00944-f005:**
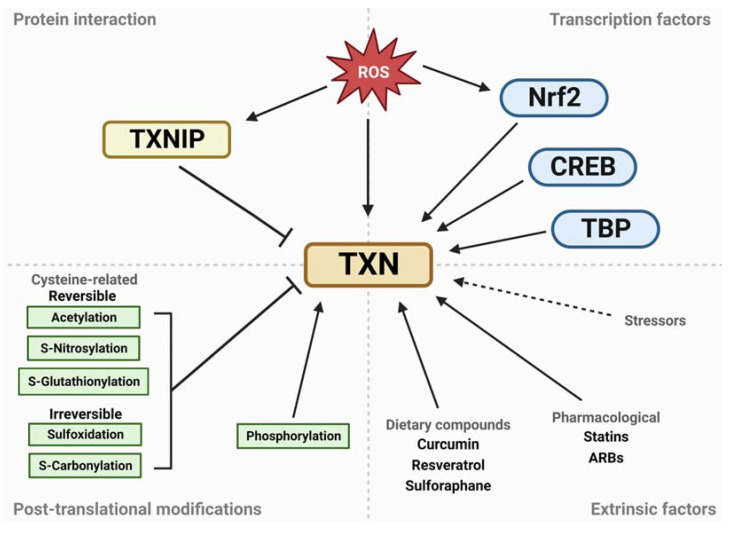
Regulation of the thioredoxin system. The regulation of the thioredoxin system occurs at multiple levels, including gene expression, post-translational modifications, protein–protein interactions and extrinsic factors. The thioredoxin system is regulated at the gene expression level by transcription factors, including Nrf2, TBP and CREB, which bind to specific cis-regulatory elements located in the promoter regions of the genes that encode thioredoxin and thioredoxin reductase. Post-translational modifications, such as phosphorylation, acetylation or S-nitrosylation, can also modulate the activity of the thioredoxin system. Protein–protein interactions are also important for the regulation of the thioredoxin system, with TXNIP being an important endogenous molecule that negatively regulates the function of thioredoxin. In addition to these intrinsic regulatory mechanisms, the thioredoxin system can also be modulated by extrinsic factors such as dietary interventions, drugs or environmental stressors. The figure shows the different regulatory mechanisms that control the activity of the thioredoxin system and their effects on its function.

## Data Availability

Not applicable.

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
