# Peer review of "Evolutionarily Conserved Role of Thioredoxin Systems in Determining Longevity"

_antioxidants, 2023, doi:10.3390/antiox12040944_

Round 1

Reviewer 1 Report

A manuscript entitled of “Evolutionarily conserved role of thioredoxin systems in determining longevity” by Abdelrahman AlOkda and Jeremy M. Van Raamsdonk, was submitted as a review for possible consideration to publish on Antioxidants. Overall, the effects or roles of thioredoxin and thioredoxin reductase were reviewed and discussed in C. elegans, Drosophila, mice, and human. The association or relation between thioredoxin systems and longevity somehow may be noted to readers. Useful Figures and informative Tables were less than enough.

Major Issues:

1)     The title was incorrect. I hardly saw any solid evidence to prove that either thioredoxin or thioredoxin reductase can paly roles in determining longevity of living creatures in particular right here for mammals.

2)     Not all thioredoxin reductases present as Sec-containing selenoproteins in nature. Thioredoxin is not the only unique physiological substrate of thioredoxin reductase. TRP-14 is TXN-like protein and function well in bodies.

3)     Clearly, the cytosolic TXNRD1 differs greatly from the mitochondrial TXNRD2. Regarding the lifespan, I am not sure that both TXNRDs can perform the same or similar function in prolong the longevity for mammals.

4)     As we can read that Figure 1 showed the role of thioredoxin system in antioxidant defense. Yes, basically, I am not very much against the Figure 1, but I am confused why authors plotted Figure 1 here. ROS may affect the lifespan, and it can be proposed. However, what’s the correlation between this figure and the conclusion of this manuscript? What’s the most determining factors of longevity in mammals, are they shown in Figure 1?

5)     Figure 2 showed the role of thioredoxin in redox signaling. Still, I am confused about the necessary for presenting Fig.2, which is a little bit far from the focus of this manuscript.

6)     As for Table 1, forgive me to say so, truly its information content was kind of low and pale.

7)     Table 2 seems to be O.K., linking the lifespan. and well presented. But, where were the mechanisms? Why disruption of thioredoxin gene results into the effects on lifespan? Why overexpression of genes improves the lifespan? Possible mechanism of regulating the lifespan in details?

8)     We should make it clear that “Cytoplasmic thioredoxin is important for longevity in C. elegans” does not mean “TXN1 is essential”. So far so difficult to say that “TXN determines the longevity in C. elegans”.

9)     The authors talked about the “Contribution of thioredoxin systems to lifespan in Drosophila”. Very good, I believe that either effects or contributions of TXN systems can be reviewed and discussed in a proper way. It could be helpful.

10)  The authors wrote that “Cytoplasmic and mitochondrial thioredoxin systems are required for survival in mice”. It seems to be much clearer that survival does not equal to longevity. To survive or go die, for mammals or cells, does not mean live longer or shorter. Please trust me, they are different concepts. Mix-up induces chaos.The authors were saying that “Thioredoxin reductase variant is associated with longevity in humans”. I may agree with such ideas but more evidences or reports should be provided to support the authors’ thoughts until the truth is fully unrevealed one day.

Reviewer 2 Report

The manuscript of AlOkda et al. is a review on the relationship between thioredoxin system and longevity. The review is very useful for the community and is very well documented. It is summarizing very well all the knowledge for this scientific question. However it could be improved a lot with more illustration (see below)

-          L42. Introduce the notation for thioredoxin and thioredoxin reductase here

-          Part 2  starting L48. A figure with the 3D structure of the protein is very useful

-          Part 2 starting L72. A figure with the mechanism of the reaction will help the readers.

-          Part 4. Starting L181. A figure summarizing the different regulations of the thioredoxin system is needed. In the title of this part replace “Regulation” by “Regulations”

-          Table 2. The signs used in the column “Effect on Lifespan” have to be defined precisely. From which variation do you consider that there is an increase or a decrease compared to no variation

Round 2

Reviewer 1 Report

The authors have made good improvement for the last version. However, the new version has major problem in the Fig 4 and 5. Both figures are centered around TXN, whereas ignored the role of TXNRD. This is not appropriate, since TXNRD is dominant for the reduction of TXN. The Fig. 4 and 5 should be further revised. I suggest acceptance after the major revision of Fig 4 and 5.

Author Response

Dear Reviewer 1,

Thank you for reviewing the revised version of our manuscript. Please see below for our response to your comments.

The authors have made good improvement for the last version. However, the new version has major problem in the Fig 4 and 5. Both figures are centered around TXN, whereas ignored the role of TXNRD. This is not appropriate, since TXNRD is dominant for the reduction of TXN. The Fig. 4 and 5 should be further revised. I suggest acceptance after the major revision of Fig 4 and 5.

We agree that thioredoxin does not act independently and that thioredoxin reductase is important for the reduction of thioreodoxin into its active form. We have described the role of thioredoxin reductase in reducing thioredoxin in Section 2 of the manuscript and illustrated this in Figure 1. In Figure 4, we have indicated that it is the active, reduced form of thioredoxin that is contributing to redox signaling. As there are multiple places in which reduced thioredoxin is illustrated in the figure, we have elected not to incorporate thioredoxin reductase into this figure as it would become overly cluttered. Instead, we have added a sentence to the Figure legend to acknowledge the role of thioredoxin reductase in reducing thioredoxin. Figure 5 is focussed on thioredoxin and not thioredoxin reductase as this is what we have described in Section 4 of the manuscript. We have chosen to focus on the regulation of thioredoxin in Section 4 of the manuscript as this is the focus of the Special Issue to which we have submitted this manuscript.   

Reviewer 2 Report

The authors answered to the different points that I rose. The paper is ready to be accepted for publication

Author Response

Thank you for reviewing our revised manuscript and indicating that it is ready for publication.